# 3D Tumor Spheroid and Organoid to Model Tumor Microenvironment for Cancer Immunotherapy

Yichen Zhu [†], Elliot Kang [†], Matthew Wilson, Taylor Basso, Evelynn Chen, Yanqi Yu * and Yan-Ruide Li *

Department of Microbiology, Immunology & Molecular Genetics, University of California, Los Angeles, CA 90095, USA
* Correspondence: yu63@g.ucla.edu (Y.Y.); charlie.li@ucla.edu (Y.-R.L.);
  Tel.: +1-812-964-9643 (Y.Y.); +1-310-254-6068 (Y.-R.L.)
† These authors contributed equally to this work.

**Abstract:** The intricate microenvironment in which malignant cells reside is essential for the progression of tumor growth. Both the physical and biochemical features of the tumor microenvironment (TME) play a critical role in promoting the differentiation, proliferation, invasion, and metastasis of cancer cells. It is therefore essential to understand how malignant cells interact and communicate with an assortment of supportive tumor-associated cells including macrophages, fibroblasts, endothelial cells, and other immune cells. To study the complex mechanisms behind cancer progression, 3D spheroid and organoid models are widely in favor because they replicate the stromal environment and multicellular structure present within an in vivo tumor. It provides more precise data about the cell–cell interactions, tumor characteristics, drug discovery, and metabolic profile of cancer cells compared to oversimplified 2D systems and unrepresentative animal models. This review provides a description of the key elements of the tumor microenvironment as well as early research using cell-line derived, 3D spheroid tumor models that paved the way for the adoption of patient-derived spheroid and organoid models. In particular, 3D spheroid and organoid models provide a method for drug screening with a particular emphasis on influence of the TME in cancer immunotherapy.

**Keywords:** 3D culture; tumor spheroid; tumor microenvironment (TME); patient-derived spheroid; patient-derived organoid; tumor-associated macrophage (TAM); immunotherapy; chimeric antigen receptor (CAR); CAR-engineered T (CAR-T) cell

## 1. Introduction: The Tumor Microenvironment as a Barrier to Cancer Immunotherapy

The tumor microenvironment (TME) is a complex and dynamic entity composed of malignant and non-malignant cells, including innate and adaptive immune cells, fibroblasts, adipocytes, vascular and lymphatic endothelial cells, pericytes, and the extracellular matrix (ECM) (Figure 1a) [1]. In the TME, immune cells fail to exert cytotoxic antitumor functions, promoting an immunosuppressive environment that facilitates tumor growth and proliferation [2]. Advancements in the understanding of cellular and molecular interactions in the TME and their impact on tumor progression may provide an alternative avenue to simultaneously and synergistically achieve tumor suppression tailored to individual patients [2–4]. The TME is widely recognized as a barrier to cancer therapy due to its diverse and pervasive roles in tumorigenesis and late-stage tumor evolution [2–6].

A combination of cellular and acellular components contributes to the overall complexity of the TME and serves as a potential target for therapeutic intervention [3]. Tumor endothelial cells (TECs), which differ from normal endothelial cells (NECs) in morphologic and genetic phenotype, play a vital role in tumor angiogenesis, forming new blood vessels and regulating the passage of nutrient factors to tumors [7,8]. TECs are morphologically characterized by irregular cell surfaces, fenestrated cell walls, and loose intercellular junctions. Furthermore, they can downregulate the expression of adhesion molecules and

chemokines to reduce immune cell trafficking [7]. Cancer-associated fibroblasts (CAFs), another cellular component of the TME, support primary tumor growth and metastasis; they not only secrete growth factors and cytokines to induce tumor growth, but also act to mutagenize pre-cancerous cells [9]. Additionally, CAFs are linked to the initiation of desmoplasia, a feature commonly displayed in tumors characterized by increased ECM deposition [10,11]. Although the ECM is normally at equilibrium between protein formation, degradation, and post translational modifications (PTM), this balance is lost during malignant transformation when the ECM is overproduced and remodeled abnormally, altering its overall biochemical properties [11,12]. The imbalanced ECM assists cancer cells in evading growth suppression, resisting cell death, sustaining proliferative signaling, enabling replicative immortality, inducing angiogenesis, and activating invasion and metastasis [12]. Moreover, the greatest attention is directed toward immune cell components–in particular, tumor-associated macrophages (TAMs) and myeloid-derived suppressor cells (MDSCs) [3,13,14]. These two cell types serve as the primary immunosuppressive force inside the TME, affecting the adaptive immune system across multiple pathways [15–19]. TAMs suppress effector immune cells through multiple methods including the expression of ligand receptors for immune checkpoint proteins, including programmed cell death protein 1 (PD-1) and cytotoxic T-lymphocyte antigen 4 (CTLA-4), and the secretion of cytokines and chemokines that activate regulatory T cell (Treg)-mediated suppression pathways [20]. These elements are subject to bi-directional stimulation; MDSCs facilitate tumor immune escape and angiogenesis, while tumor cells secrete growth factors and chemokines that induce MDSC proliferation [17]. Previous studies have shown that nuclear factor $\kappa$B (NF-$\kappa$B) signaling pathway can be specifically inhibited to restore TAM cytotoxicity, and that hypoxia-inducible factor-1$\alpha$ (HIF-1$\alpha$) deletion in macrophages can reduce tumor growth [15,16]. Tumor trafficking of MDSCs is also inhibited by CXCR2 deficiency or anti-CXCR2 monoclonal antibody therapy, increasing the efficacy of anti PD-1 therapy through targeting of the CXCL5-CXCR2 signaling pathway [18,19,21].

With the complexity of cell types and ECM inside the TME, solid tumors are difficult to target due to the biological and mechanical hindrance of drug delivery [22–25]. One previous study found that an abnormally-formed vascular system inside the TME led to interstitial hypertension, heterogeneous blood supply, and relatively long transport distances in the interstitium (Figure 1b) [22]. For example, doxorubicin, a commonly-used cytotoxic agent, is unable to penetrate more than 40–50 μm from blood vessels to sufficiently target tumor cells [26,27]. Chemotherapy penetration rate through tumor tissue may be diminished to as low as 20% due to the presence of a multicellular layer (MCL) within the ECM containing laminin and collagen [23]. The poor penetration capability of chemical drugs through the MCL contributes to their low efficacy through solid tissue tumor models [23,24]. In addition, hypoxic conditions of lower than 5–10 mmHg in the TME is responsible for TAM, Treg, and MDSC accumulation in affected regions, contributing to a more aggressive and therapeutically-resistant tumor phenotype [28–32]. Hypoxia also induces a rapid, dramatic, and selective upregulation of programmed cell death ligand-1 (PD-L1) in multiple cancer types on MDSCs, macrophages, dendritic cells, and tumor cells via HIF-1$\alpha$ pathway [16,33,34]. A blockade of PD-L1, the ligand of PD-1 immune checkpoint receptor, has been observed to reduce MDSC-mediated T cell suppression under hypoxic conditions [34]. Stress-induced or cytotoxic drug-induced autophagy, which may be driven by hypoxia, also significantly enhances tumor cell survival inside the TME [35]. Inhibition of tumor suppressor gene aplasia Ras homolog member I (ARHI)-induced autophagy successfully reduced the regrowth of xenografted tumors, indicating the role of autophagy within tumor dormancy [36]. It was also determined that the degradation of granzyme via hypoxic tumor cell autophagy suppresses natural killer (NK) cell-mediated apoptosis, consequently increasing immune escape [37].

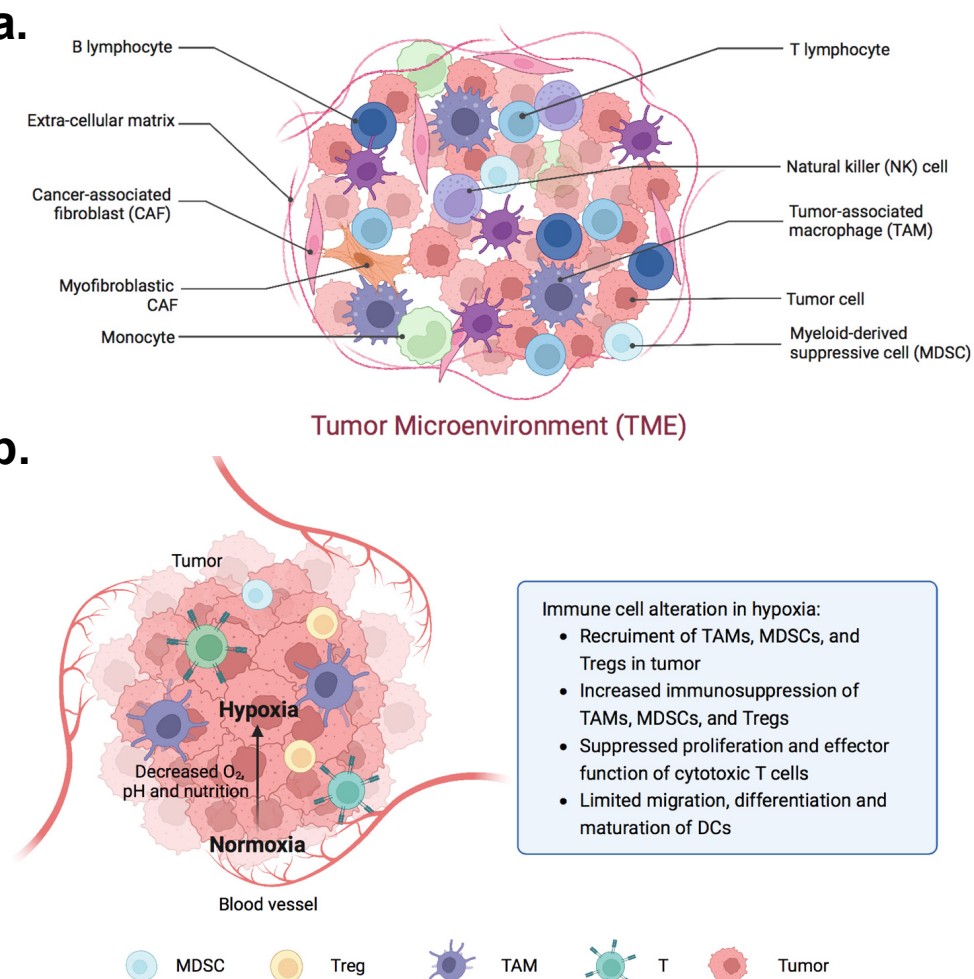

**Figure 1.** Schematic representations of the immunosuppressive TME. (**a**). In a solid tumor microenvironment, tumor cells coexist with tumor associated-macrophages, myeloid-derived suppressive cell, fibroblasts, and tumor-infiltrating lymphocytes. (**b**). Hypoxia is a key component of the tumor microenvironment and is tightly correlated with immunosuppressive cell recruitment, tumor immune response, angiogenesis, metabolism, and cell proliferation.

Nanoparticle drug delivery systems may potentially overcome the biological and mechanical barriers involved with current cancer therapeutics for improved treatment outcomes [38–40]. The delivery platform not only works for chemotherapy, but also assists nucleic acid therapy development by providing liposomes, magnetic nanoparticles, and gold nanoparticles as the vehicle for gene delivery [38,41]. In immunotherapy, nanoparticles may be modified with ligands to specifically target different components in the TME, including dendritic cells, macrophages, and fibroblasts [39]. Nevertheless, there remains several challenges with the implementation of nanomedicine into clinical practice. Factors such as particle size, particle shape, surface charge, deformability, and degradability must be carefully considered for adequate delivery of therapeutics to tumors [38]. Additionally, the complex immune network during tumorigenesis inside the TME remains unknown, which may elicit unpredictable immune responses to nanoparticles and calls into question their implementation [39].

Only 5% of anticancer drugs enter the clinical stage due to inappropriate pharmacokinetics or lack of drug efficacy [42]. More specifically, a drug that successfully targets a specific cell type in vitro may be rendered ineffective by other components in the TME [43]. Therefore, new methods are necessary to acquire more effective drug delivery outcomes. As difficulties in the successful translation of novel therapeutics are mainly attributed to the complicated nature of the TME, as is discussed above, a better understanding of

the processes governing these barriers during the early stages of therapy development is essential to maximize drug delivery success rate.

## 2. 3D Modeling of the TME

Most research in cancer biology is based on experiments involving 2D cultures; however, 2D cell culture techniques have many limitations. Direct comparisons indicate that the TME is better simulated through 3D cultures compared to 2D tumor models, as 2D methods fail to mimic cell–cell and cell–extracellular environment interactions [44–46]. Additionally, the modes of cell division and adhesion are limited under 2D conditions [44]. As the traditional monolayer cell culture model cannot accurately test drug resistance and may display misleading results, efforts have been directed toward the field of new in vitro tumor models to better represent the TME [43,44,46–48]. Modern approaches of multicellular layers, 3D scaffolds, 3D bioprinting, microfluidics, and spheroid models were established to model the TME through relatively inexpensive and convenient methods [43,44,47–54].

In the last century, multicellular layers (MCLs, also referred to as multicellular models or MM models) were developed. This in vitro model mimics the TME and is based on culturing cells at tissue-like densities on collagen-coated microporous Teflon membranes [49]. Since cells cultured via MCLs are physiologically similar to cells observed in in vivo tumors, researchers have used them to study the penetration of chemotherapy anticancer drugs [24,49]. In addition, 3D bioprinting–a novel technology–can be further utilized to construct a 3D scaffold, preserving the tumor network structure and ECM architecture, while manipulating the microenvironment with high reproducibility [50]. Microfluidic cell culture, in contrast, can independently modify all parameters of the synthetic culture system, such as cell types, cell positions, and precise orientation of tissue–tissue interfaces [52]. Additionally, the Tumor-on-a-Chip Device allows the interaction study among breast cancer cells, monocytes, and endothelial cells to examine T-cell infiltration [53].

While engineering-based approaches emphasize model structure and composition, cell-based approaches utilize the inherent capabilities of cells to organize into 3D aggregates without many external cues [54]. Three-dimensional spheroids were initially developed as a more cost-effective and simple approach to 3D modeling compared to prior attempts; this technology has been implemented to facilitate the production of cell spheroids under reproducible conditions in a high-throughput manner [47,54]. The primary features found in solid tumors in vivo and in other 3D models, such as cellular heterogeneity, cell–cell signaling, ECM interactions, growth kinetics, and drug resistance, can all be successfully replicated in 3D spheroids [47]. Further advances using patient-derived tissues have led to the development of organoid models that mimic the structural heterogeneity of the TME with even greater precision [55]. Moreover, comparable in vivo models face certain limitations, including high costs, high labor intensity, and the lack of functional immune systems in xenograft mice models [54,56]. Therefore, 3D spheroids and organoids can complement current therapeutic development strategies, filling in the gap between in vitro and in vivo research.

## 3. Cell Line-Derived Tumor Spheroids

### 3.1. Culture Methods

A variety of methods have been implemented to assemble tumor-derived spheroids; both anchorage-independent and -dependent models have been explored to varying degrees of success [57]. An experiment conducted by Al-Hity et al. utilized ultra-low attachment 96-well plates to culture spheroids formed from murine breast cancer cell lines (i.e., either 66CL4 or 4T1) and verified their integrity and fidelity through structural comparison to in vivo samples [58]. Not only was their architecture extremely similar but they also displayed similar vulnerability to immune cell infiltration [58]. Other strategies taking advantage of cancer cells' ability to spontaneously aggregate in nonadherent conditions include the magnetic levitation, hanging drop, and spinner flask methods [48,59]. However, as historically observed, ultra-low attachment substrate highly restricts the number of

spheroids formed, provides inconsistent development, and creates variation in spheroid ECM content, constraining testing consistency [60]. Nevertheless, the ability to assemble spheroids provides an avenue for a wide array of testing, including drug sensitivity and penetration, immune cell infiltration, and tumor architecture.

To overcome these limitations, variations on standard spheroid formation have implemented various scaffolding types to control spheroid structure [61]. For example, successful cultures have been accomplished using ultra-porous cellulose for PC3 prostate cancer epithelial cells [62], fibrous scaffolding for MCF-7 breast cancer cell line [63], and alginate encapsulation for the NCI-H157 NSCLC cell line [59]. The adoption of biocompatible scaffolds to support spheroid development has produced greater reproducibility, titration of physical and chemical conditions, and nutrient factor transport that contributed to greater spheroid throughput and control [64]. There are still some limitations overshadowing scaffold use; for instance, porous scaffolds have restrictions on their ability to diffuse nutrients and fibrous scaffolds do not completely represent a 3D ECM due to promoting cell growth along fiber strands [61,64].

Besides scaffolding, another four approaches have been utilized for spheroid formation in a scaffold-free culture [65]. In agitation-based methods, cells are kept in stirring condition to avoid cell adhesion to surfaces for enhanced spheroid aggregation [65]. The hanging drop technique grows spheroids within drops of culture medium, taking advantage of the surface tension of liquid [65,66]. In the liquid overlay technique, non-adhesive surfaces such as agarose are seeded with cells for spheroid formation without cell attachment [67]. In contrast, the microfluidic system has cell culture microchambers and hemispherical microwells with a concentration gradient generator [68]. The complex system provides controlled mixing, chemical concentration gradients, lower reagent consumption, continuous perfusion and precise control of pressure and shear stress on cells [69]. It was reported by Ruppen et al. that spheroids formed with continuous perfusion of drugs would have higher drug resistivity [70]. With some further enhancement, long-term monitoring and high-throughput drug screening platforms could also been achieved through the microfluidic system [68,71].

*3.2. TME Modeling Capabilities of Spheroids*

As metabolic products accumulate (e.g., lactate), acidic environments in tumors have been well observed in the interstitial space and ECM, displaying a heterogeneous acid—base phenotype [72]. Acidic conditions inside the TME, which may subside to as low as a pH of 5.6, contribute to drug resistance for both chemotherapy and radiotherapy [65,72,73]. As shown by Carlsson and Acker in 1988, 3D spheroids similarly display a low pH within their deepest regions, indicating their potential usage as a drug resistant screening platform [74]. In a separate study by Swietach et al., HCT116 cells were cultured as multi-cellular 3D spheroids. A radial pH gradient was established with the lowest level at the core, as the extracellular space restricts the diffusion of metabolically produced acids [75]. With the assistance of spheroids, the efficacy of doxorubicin—a weakly basic drug—was found to be pH-dependent in the TME due to its reduction upon drug entry and drug accumulation [75]. The protonated form of doxorubicin crosses cell membranes slower than the unprotonated form, implying that other basic chemotherapeutic drugs may also be hindered by the acidic environment [75]. Moreover, 3D spheroids were utilized to study the role of Carbonic anhydrase IX, a hypoxia-inducible tumor-associated cell surface enzyme, in the TME acidification procedure [76].

The development of hypoxic conditions from high lactate production and poor tumor vascularization usually occupies an overlapping gradient similar to the acidic microenvironment discussed above [77,78]. Since some anticancer drugs require oxygen to exert their antitumor effects, hypoxia, in combination with acidosis, reduces the formation of reactive oxygen species (ROS) that target cancer cells [79]. Although hypoxic regions can be modeled in 2D cell cultures with a hypoxic chamber, physiological gradients of the TME hypoxia are better simulated by 3D spheroids through the replication of an anoxic

core [78]. Therefore, the generation of 3D spheroids produced an effective method to screen for compounds that could target the TME's hypoxic phenotype [78,80,81]. The 3D, high-content screening platform proposed by Wenzel et al. enables an efficient identification procedure of compounds that may induce cell death specifically in inner spheroid regions with low oxygen availability [80]. This platform provides an advantage over traditional 2D-based screening for the identification of substances that would be otherwise overlooked in the absence of the hypoxic gradient [80]. In another study, nanoimprinting 3D spheroids, together with cancer cells extracted from patient tumors, indicated a similar formation of hypoxic regions in comparison to in vivo tumors, further enhancing the reliability of the 3D screening platform [81].

Due to hypoxic conditions in the TME, tumor cells generate energy through an alternative metabolic pathway than untransformed cells, which standardly undergo mitochondrial oxidative phosphorylation [65,79]. The anaerobic metabolism pathway, instead of the normal tricarboxylic acid (TCA) cycle, was first observed in carcinoma cells by Otto Warburg in 1925 [82]. Accordingly, glucose is utilized to produce lactate as a by-product inside the TME, resulting in lactate accumulation and pH depression [65,79]. In 3D spheroids, Khaitan et al. discovered that glucose consumption and lactate production were much higher in spheroid cells than in monolayer cells. Meanwhile, ROS levels were largely reduced in the spheroids with increasing age and size. These features in 3D spheroids allow researchers to evaluate tumor response to metabolic inhibitors such as 2-deoxy-D-glucose (2-DG), an inhibitor of glucose transport and glycolysis [83]. Moreover, due to their similar metabolism environment as the TME, 3D spheroids were used to study amino acid (i.e., leucine and glutamine) function in melanoma tumor growth, and to map lipid distribution and profile in breast cancer progression [84,85].

As described above, the TME is characterized by high heterogeneity, consisting of multiple cell types, and displaying various kinds of cell–cell interactions. In addition, 3D spheroid models can culture different cell types together to generate multicellular spheroids, mimicking the cellular heterogeneity of solid tumors and tumor-stromal cell interactions [65]. Tumor cells, fibroblasts, endothelial cells, and immunocompetent cells have all been cocultured in the spheroid generation process [86]. Early attempts co-cultured a single cell line tumor spheroid with polarized TAMs, demonstrating TAM migration and spheroid infiltration [87,88]. Further modeling has shown that TAM migration plays a strong role in tumor pathogenesis by promoting tumor infiltration into the surrounding matrix, having a mechanistic role in tumor progression and metastasis [89]. More sophisticated approximations using multicellular culturing have been accomplished through the generation of 3D spheroids composed of a mixture of tumor cell line, CAFs, and monocytes with appropriate nutrient factors [90]. Interestingly, the 3D TME spheroid captured the de novo polarization of naïve monocytes towards a TAM immunosuppressive phenotype, displaying the fidelity under which multicellular spheroid model operates [90].

Contributing to the success of 3D spheroids, cell–cell interactions can be more clearly observed in 3D spheroids than in 2D cultures, influencing cancer cell signaling, survival, proliferation, and drug sensitivity [65]. For example, Xu et al. demonstrated that E-cadherin, as one of the widely known adhesion receptors, would increase its expression in the multicellular spheroids of ovarian cancer cells [91]. These results indicate that E-cadherin induces spheroid formation, maintenance, and drug resistance in ovarian cancer [91]. Consistently, the administration of an E-cadherin blocking antibody on spheroids led to improved cellular death in liver cells [92]. Another vital feature in solid tumors is the uncontrolled ECM, a noncellular component present in all tissues composed primarily of structural proteins, nonstructural proteins, and other components such as growth factors and Matrix Metalloproteinases (MMPs) [93]. The dysregulation of the ECM's composition, structure, stiffness, and abundance contributes to fibrosis or cancer pathogenesis [94]. Nederman et al. studied the ECM proteins in spheroids of a human glioma cell line (U-118 MG) and a human thyroid cancer cell line (HTh-7) and found that 3D spheroids show increased expression of ECM proteins, including fibronectin, laminin, and collagens [95].

Notably, spheroids may be further engineered to better capitulate the ECM feature of the TME [96,97]. The ECM amount (thickness) between cells was modulated by Tao et al. by replenishing ECM components and polysaccharides into spheroids with the maintenance of cell viability, enabling a more precise fabrication of spheroid models [96].

Cancer stem cells (CSCs) are defined as a fraction of cancer cells capable of generating entire cancer structures due to their self-renewal and differentiation potential [98]. As the formation of spheroids in vitro is regarded as a convenient marker to identify CSCs, tumor-derived spheroids exhibit a close relationship to CSC research [98]. Breast cancer spheroids were widely used to study CSCs and their response to chemotherapies [99,100]. Three in vitro models (a 3D collagen embedded multicellular spheroid tumor model; a 3D collagen model with a single cell-type diffusely embedded; and a 2D monolayer) were compared directly to culture two breast cancer cell lines. As expected, only the 3D spheroid model displayed the enriched CSC content and high chemoresistance observed in in vivo models [99]. It was also reported in a different study that CSCs lose their drug resistance when grown in monolayers. However, CSCs and cells grown in spheroids are highly resistant to chemotherapeutic agents [100]. These comparisons clearly indicate that 3D spheroids provide the best model to study CSC populations.

The internal structure of spheroids contains an external layer of highly proliferative cells, a middle layer of senescent cells, and a core of necrotic cells [47,101]. In combination with the features of hypoxic regions, increased ECM mass, and close cell–cell interactions, the architecture of spheroids forms a physical barrier that limits the penetration and delivery of drug compounds [65]. This resistance has been well observed and defined above.

Despite their large similarity to the real TME, 3D spheroid models continue to have certain limitations [54,66]. The lack of control on the spheroid architecture may lead to the formation of heterospheroids, which have an opposite hierarchy compared to the solid tumor [54]. In addition, spheroid formation is influenced by cell type, culture technique, medium composition and volume, and cell density, which could introduce variability in production [66]. Further, the culturing and growth time required in spheroid formation is lengthy, which may be improved with the help of novel technologies such as microfluidic chips [66].

## 4. Patient-Derived Spheroid and Organoid Models

Although cell-line-derived spheroid models are in many cases sufficient in modeling the TME, further granularity is enabled by the creation of 3D models from primary tumor tissue. Cell lines often have genetic modifications introduced during the immortalization process, such as the introduction of simian vacuolating virus 40 (SV40) large T antigen, which may alter the expression profile and biology of 3D models utilizing them [55]. In other cases, cell lines may be isolated as clonal outgrowths during culture crisis, causing the overrepresentation of a specific culture-adapted phenotype [102]. Patient-derived models seek to overcome these limitations by directly culturing tumor samples, preserving the intratumoral heterogeneity integral to accurate modeling of the TME. Patient-derived xenografts (PDX) are an already widely implemented patient-derived model and are commonly used in preclinical studies for novel anticancer drugs. However, PDXs come with their limitations as they have been shown to be susceptible to infiltration by murine tissue, including myeloid cells [103]. Furthermore, they are slow-growing and consequently low-throughput [103,104], which makes them non-ideal for drug screens, especially as combinatorial studies requiring higher throughput become more commonplace [105]. An additional consideration is the ethical issues surrounding the use of PDXs.

Developed using pervious work on cell-line spheroids, patient-derived spheroid/organoid models circumvent risks associated with using PDXs while also retaining the advantage of using primary tissue. They have been shown to accurately model the transcriptional profile of source tumors, and certain formation methods are able to preserve the cellular heterogeneity of the parent tumor, adding another dimension of accuracy to the modeling of the tumor. In addition, they are amenable to high-throughput screens and have

been successfully integrated with automated systems. Some have been shown to survive cryopreservation [106], creating another dimension of collaborative research with the establishment of tissue banks [107]. Although not without its limitations, patient-derived spheroids/organoids have been a rising model of choice for researchers characterizing and modeling solid tumors [108].

Unfortunately, the classification of patient-derived 3D in vitro models has historically been inconsistent, complicated by the wide variety of formation methods established in literature [108,109]. A broad system is employed for the purposes of this review: patient-derived models grown without an externally provided scaffold will be classified as "spheroids," and are analogous to cell-line spheroids in that they do not have extensive structural heterogeneity. Most patient-derived models created within a scaffold have been shown to self-assemble into organlike structures, and these models will be classified as "organoids" [110,111]. It is critical to note that the constantly evolving landscape of 3D cell culture entails that there will be exceptions to this schema. Additionally, models in each of these categories can be further differentiated by the source of the cultured cells, which can vary widely [55].

### 4.1. Patient-Derived Spheroids

The techniques used to culture patient-derived spheroids are similar to those utilized for those made from cell lines. Ultra-low attachment plates, spinner flasks, and hanging-drop methods have been successfully used on cells either mechanically or enzymatically digested into suspension [43,112]. In all these models, the nonadherent conditions result in the inter-cell adhesion to be stronger than the adhesion to a 2D surface, resulting in the spontaneous formation of a 3D aggregation of cells. Colorectal cancer spheroids and breast cancer PDX spheroids, referred to as "colospheres" and "MARY-X spheroids", respectively, were shown to spontaneously form in suspension after the initial mechanical dissolution of the tumor [109,113,114]. Notably, this property was unique to colorectal cancer samples, and nonneoplastic tissue did not form colospheres [115]. More recently, chordoma spheroids were formed in ultra-low attachment wells from digested primary chordoma tissue [116], and similar methods were employed for the culture of ovarian and bladder cancer [117,118]. A hanging-drop protocol that is conducive to high-throughput screens in a 384-well format has also been described [111].

Limited information is available on TME modeling specifically for patient-derived spheroids, but similarly to cell-line spheroids, they accurately model cell–cell ECM interactions, gene expression, acidity, hypoxic gradient, and the necrotic core—particularly when compared to 2D models [43,110]. Similarly to cell-line models, coculture with CAFs are able to further recapitulate the heterogeneity of the parent tumor, as demonstrated in multiple breast cancer subtypes [119].

As mentioned above, a unique ability of 3D models in low-attachment conditions is their capacity to enrich cells exhibiting a cancer stem cell (CSC)-like phenotype [109]. This has been exploited to study signaling pathways in bladder cancer CSCs, resulting in the characterization of aldehyde dehydrogenase 1A1 (ALDH1A1) and tubulin beta III (TUBB3) as potential biomarkers and treatment targets [118]. Furthermore, patient-derived spheroids are particularly relevant in the study of ovarian cancer as the higher level of CSCs is reflective of metastatic ascites in the late stages of the disease [111]. In a related study, cells derived from ascites of high-grade serous ovarian cancer patients were used to culture monoclonal spheroids. Increasing passage numbers were found to augment spheroid formation efficacy, indicating an enrichment of CSCs [120].

In a special case, spheroids formed on low-attachment plates from digested nondesert ovarian cancer tissue were able to retain the patients' immune cells in vitro [117]. CD8+ T cells, Tregs, and DC populations were detected, as well as macrophage inflammatory protein-1 alpha (MIP-1$\alpha$) and tumor necrosis factor alpha (TNF$\alpha$), which implied the presence of macrophages. Although the formation of spheroids disrupted the relative abundances EpCAM+/PD-L1+ tumor cells and Tregs, this study indicates that the culture

of patients' TILs is possible, which adds a further degree of accuracy compared to coculture with peripheral leukocytes [117].

Although patient-derived spheroids are less commonly used than patient-derived organoids, they have been demonstrated to be useful in the modeling of drug treatments. Compared to organoids, which require an ECM substitute to be provided, spheroids are more cost-efficient and easier to handle [119]. This also allows for convenient imaging and downstream analysis due to the lack of a matrix around the aggregate [55,119]. Due to this efficiency, high-throughput screens can be made by simply using a 384-well hanging drop method. This has the additional advantage of overcoming any intra-patient heterogeneity issues through the use of multiple replicates [111]. Treatment of breast cancer MARY-X spheroids with various anticancer drugs resulted in the validation of the caged Garcinia xanthone (CGX) motif as a potential therapeutic. Other common therapeutics were ineffective, highlighting the application of patient-derived spheroids in screening for novel drugs [114]. In the realm of cancer immunotherapy, B7-H3-CAR-T cells were shown to be effective against skull base chordoma spheroids. Notably, patient-derived spheroids were used instead of PDX due to an inability to form a PDX model of skull base chordoma [116]. The ovarian cancer model containing patient TILs was shown to respond to PD-1/PL-L1 blockade in combination with olaparib, demonstrating the activity of endogenous lymphocytes in vitro [117].

Patient-derived spheroids exhibit the limitations common to patient-derived in vitro models, and their variable establishment rate can sometimes hinder the feasibility of drug screens [110]. Notably, a significant downside specific to spheroids is their lack of a properly defined ECM [111]. This eliminates a major component of the TME and simultaneously prevents cells from self-organizing in a structure mimetic of in vivo tumors [121]. To properly model these systems, patient-derived organoids are necessary [117].

### 4.2. Patient-Derived Organoids

Single-cell suspensions or minced chunks embedded in ECM substitutes such as Matrigel or Geltrex have the capacity to self-organize into 3D structures that resemble the parent tissue [122]. The modern development in cancer modeling through patient-derived organoids (PDOs) is based off of the work of Sato et al. and has since led to the successful culture of colon [102], pancreas, appendiceal, prostate, breast, gastric, cerebral, lung, esophageal, bladder, ovarian, kidney, and liver tumor tissue [43,123].

Most patient-derived organoids follow the culture method established by Sato et al. [102,104]. In brief, cells are suspended in a mixture containing an ECM substitute, which then solidifies at higher temperatures (Figure 2a). Organoids can be passaged by mechanical or enzymatic digestion and reseeding into a new matrix [102]. The properties of matrix itself can modulated by using different mixtures or with the use of an "absorber" step to enhance rigidity [55,124].

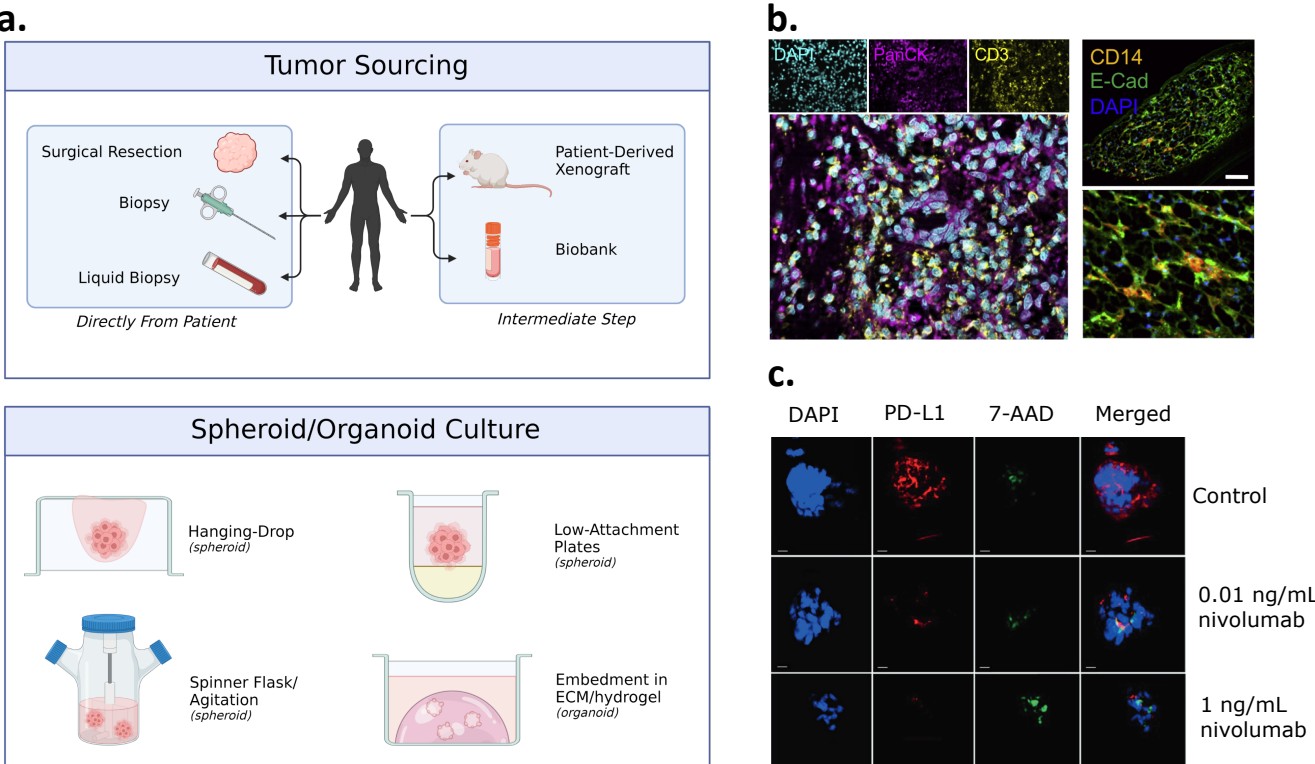

**Figure 2.** Schematic representations of the culture method of patient-derived spheroids and organoids. (**a**). Visualization of patient-derived spheroid/organoid formation workflow. (**b**). Representative example of TME modeling through the preservation of primary immune cells in an organoid culture. IF staining of clear cell renal cell carcinoma organoid culture demonstrating the presence of TILs and TAM. (**c**). Representative example of assessing immunotherapies through patient-derived organoid culture. Treatment of chordoma organoids with nivolumab demonstrating enhanced killing by primary TILs. Note that (**b**,**c**) are reproduced from Refs. [125,126] with permission from Elsevier, copyright (2018) and Springer Nature, copyright (2019) [125,126].

Despite the similar culture methods, there is diversity in the source of the tumor cells, which can have implications in the accuracy of TME modeling. Tissue samples from surgical resection were used in the original organoid experiments and continue to be widely used as source material [102,121,124,127–129]. Unfortunately, this is not the standard of care for patients with metastatic cancer, and although the establishment of organoids from biopsies is possible [128,130,131], they have been demonstrated to be a challenging source due to a limited number of cells [104,130,132]. A potential alternative is the use of circulating tumor cells (CTC) as a noninvasive source of primary tumor cells for organoid culture [106]. This has the added benefit of sampling from multiple sites, negating the effect of intra-patient heterogeneity common in metastatic cancer [104]. An alternative method of enhancing efficacy is the derivation of organoids from preestablished PDX models. Due to the creation of "living biobanks", these have the potential to be a consistent source of patient-derived cells for organoid culture. However, establishing organoids from PDX often requires an additional processing step to remove contamination by murine cells [129,133].

Variations on the standard ECM method of culturing PDOs have also been described. A culture of original TIL and CAF from the primary tissue was achieved by raising the Matrigel-embedded tumor organoids to an air-liquid interface (Figure 2b) [125]. Coculture with CAF in a monolayer or integrated into Matrigel were used to study the effect of fibroblasts [121,134]. In a group of markedly advanced models, tumor cells were seeded onto previously-grown brain, skin, and breast organoids to model invasion and tumor–

stroma interactions [135–137]. Extensions of this schema include the use of a dorsal root ganglion explant to further improve accuracy of interactions [55].

The ability of PDOs to recapitulate the TME in vitro has been well-documented [127]. Alongside the standard benefits of 3D modeling [111], organoids have the unique ability to mimic in vivo structures [43]. Epithelial tumor cells grown in Matrigel acquire an apicobasal polarity and self-assemble into organoids not seen in cell lines [121]. Additionally, the presence of an ECM allows for further flexibility in the modeling of intratumor heterogeneity. CAFs have been shown to remodel externally provided ECM, and are able to aid invasion in scaffold-based cultures by "leading" tumor cells through the matrix [55]. Fibroblasts cocultured with pancreatic cancer organoids expressed smooth muscle actin (αSMA), the well-established marker for CAFs [121]. PDOs' accurate expression profiles when compared to the parent tumor make them ideal models for biomarker research [106].

PDOs are also relevant in modeling the TME for immunological research. Due to PDO's ability to accurately mimic in vivo conditions, coculture experiments with PBMCs have resulted in the expansion of tumor-specific cytotoxic T cells in NSCLC and dMMR CRC [128]. TCRs generated in a similar method were shown to confer tumor cytotoxicity when expressed in heterologous T cells [138]. Invasion can also be modeled by coculture of organoids and T cells, and this is particularly relevant as the ECM and CAFs can serve as barriers to the activity of lymphocytes [121]. Of particular note is the ability of the air-liquid interface model from nondigested tumor samples to retain the original immune cell populations (Figure 2b). This model is rather unique among PDO models, which typically integrate immune cells post-establishment. Furthermore, it has significant relevance in the study of immune checkpoint inhibitors, which act on endogenous immune cells [125].

In the context of cancer research, organoids are useful tools to model the effect of anti-cancer drugs and have been used extensively in drug screens, often against a wide range of therapeutics [127]. As they often model tumor expression profiles more accurately than cell lines, the efficacy of treatments with novel targets is best performed with PDOs. Conventional prostate cancer cell lines do not express androgen receptor (AR), a key factor in the progression of the disease. Testing the efficacy of bromodomain and extra-terminal (BET) inhibitors was thus only possible in PDOs that accurately preserved AR signaling [139]. PDO's complex modeling of the TME can also be beneficial in testing the efficacy of drugs, as a coculture with CAFs was shown to confer resistance to gemcitabine [108].

PDOs also play a special role in cancer immunotherapy research. Modeling of the TME is critical as cell–cell interactions can have a major modulatory effect on cell-based therapies [140]. Furthermore, the low efficacy of immunotherapies on many patients can be investigated by monitoring the patient-specific response within an organoid [128,131]. There are multiple novel cell therapies, including the expression of CCR2b on CAR-T cells [141], a cysteine-based CAR clustering mechanism [142], enrichment for CD39+ cytotoxic T cells [131], and an anti-FRIZZLED receptor CAR on NK cells [134]. In these representative studies, patient-derived organoids were particularly useful as CAR-T cells could be generated from autologous leukocytes, mimicking the clinical applications of these new technologies. Finally, the efficacy of immune checkpoint inhibitors can be investigated on PDO models that preserve TIL from the original specimen (Figure 2c) [125,126]. The accuracy of these models is further enhanced by the ECM scaffold, as the diffusion of antibodies through organoids reflects that of in vivo [108].

Although personalized medicine often refers to using genomic data to predict patient responses to therapies, functional personalized medicine can be achieved by using PDOs [133]. The development of high-throughput automated systems for testing various therapies on PDOs could direct an evidence-based approach to cancer treatment [104,106,143]. This is particularly useful in combination therapies, which have been demonstrated as especially effective, and could be efficiently screened against a patients' own specific phenotype [105]. The predictive ability of PDO drug screens to actual patient outcome has been investigated, and promising preliminary results serve as a testament to organoids; ability to accurately mimic in vivo conditions [130,133]. However, there are

barriers to the clinical use of this system, many of which relate to the low efficacy and inconsistency of establishing organoids [104].

The primary limitation for the scaffold-based PDO model is the dependence on external ECM [112]. The hydrogels create an upper limit on the density of the organoid culture, and the digestion process necessary to harvest the cells could cause damage or disrupt results [111]. Furthermore, commonly used ECM matrices, including Matrigel and Geltrex, are derived from murine cells [43,108]. This may result in an unwanted immune response against the matrix itself, as discovered with CD4$^+$ cells cocultured with PDOs [128]. A potential mitigation strategy involves using decellularized ECM (dECM) from the primary tumor, but the process is complex and lowers efficiency [43,108]. Finally, as with most patient-derived models, the success rate of establishing organoids is inconsistent, which may prevent the research and clinical use of these models [104,130].

## 5. Concluding Remarks

Increasing evidence suggests that the TME plays a significant role in promoting tumor progression, immunological evasion, and resistance to various standard-of-care therapies. Strategies targeting components of TME have emerged as promising approaches for cancer treatment in recent years, especially for solid tumors. An adequate preclinical model that resembles the TME is required for the discovery and assessment of such cancer therapeutics. With that purpose, 3D spheroids and organoids have been gaining popularity (Table 1).

**Table 1.** Comparison of the advantages and disadvantages between spheroids and organoids.

| Spheroids or Organoids | Culture Methods | Advantages | Disadvantages |
|---|---|---|---|
| Cell-line spheroids | Ultra-low attachment plates [59], Agitation-based method, Liquid overlay, Hanging drop, Microfluidics [65] | Low cost, Capability to model hypoxia and metabolic environment, Mimic cell heterogeneity and cell–cell interaction, Platform for drug screening [47] | Variability in production, Lengthy formation time [66], Lack of control on architecture [54] |
| Patient-derived spheroids | Identical to cell-line spheroids [43,144] | Accurate modeling of in vivo gene expression [43,110], Enrichment for CSCs [109], Easy handling [119] | Inconsistent formation efficiency [110], Lack of properly defined ECM [111] |
| Patient-derived organoids | Embedment of primary tissue into ECM/hydrogel [4,102,123] | Apicobasal polarity [121], Organlike heterogeneity [43], Defined ECM [55,121] | Cell density limit [111], Unwanted interaction with antigens present in matrix [43,108] |

3D spheroids and organoids enable the replication of an in vivo-like tumor microenvironment within in vitro settings, where the biochemical and physical properties of the TME including tumor hypoxia, nutrient depletion, acidosis and heterogenous gene expression are largely retained [145]. Recent developments in patient-derived spheroids and organoids have further improved the reliability of preclinical drug-screening for cancer patients. Optimally, the TME within patient-derived spheroids and organoids would contain an entire diversity of immune suppressive cells, including TAMs, MDSCs, Tregs, tumor-associated DCs and other innate immune cells, and might conceivably incorporate tumor-infiltrating immune cell populations.

Despite the dramatic growth in the usage of tumor spheroids in the in vitro evaluation of the efficacy of cancer therapies, tumor spheroids have their limitations. First, production techniques for multicellular tumor spheroids must be optimized to increase the uniformity and reproducibility of multicellular tumor spheroids. It has proven difficult to establish high-throughput drug screening platforms with heterogeneous multicellular tumor spheroids, where variabilities in size and density of spheroids would significantly distort the evaluation results. Second, the used of murine-derived ECM (e.g., Matrigel and

Geltrex), which is required for the culture of patient-derived organoids, induces external factors can influence test results. For example, matrices generated from murine sarcoma contains great amount of murine antigen that can activate immune cell subtypes such as CD4[+] T cells [128]. This constraint can be overcome through the use of synthetic or decellularized extracellular matrix. Moreover, a significant disadvantage of using tumor spheroids in preclinical investigations is the absence of standardized assays for spheroids imaging, analysis, quantification, and automation for drug screening. For instance, the most widely used fluorescence microscopy method, confocal microscopy, has a low penetration depth that prevents it from imaging large tumor spheroids [146]. However, this could be mitigated by recent advances in various noninvasive imaging methods such as electrical impedance, which has the ability to image larger spheroids using their electrical properties [144,147–149]. Analysis of spheroid characteristics such as viability and invasiveness could also be expedited with the integration of automation and artificial intelligence. Finally, the lack of supportive stroma and blood vessels is one of the intrinsic restrictions of tumor spheroid systems. To address this issue, effort has been made to develop micro-vascularized tumor organoids-on-chips [150].

In conclusion, the ability to model the TME in vitro using tumor spheroid will largely benefit therapeutic development, such as in vitro screening and optimizing pharmacological therapies. It also provides a platform to elucidate molecular pathways related to solid tumor malignancy. Although some limitation remains, recent advances in for example, 3D bioprinting, "tumor on a chip", and other microfabrication technologies will accelerate the development of more biologically relevant in vitro TME models. Ultimately, one can anticipate that tumor spheroids will be the gold standard in vitro model with a high translational predictive value, lowering the number of in vivo investigations that need to be performed, hence expediting the drug discovery process.

**Author Contributions:** Conceptualization, Y.Y. and Y.-R.L.; Writing-original draft preparation, Y.Y., Y.Z., E.K. and T.B.; writing-review and editing, Y.Y., M.W. and E.C.; Visualization, Y.-R.L. and E.K.; validation, Y.Y. and Y.-R.L.; supervision, Y.Y. All authors have read and agreed to the published version of the manuscript.

**Funding:** This research received no external funding.

**Institutional Review Board Statement:** Not applicable.

**Informed Consent Statement:** Not applicable.

**Data Availability Statement:** Not applicable.

**Acknowledgments:** We thank Lili Yang (UCLA) for her insightful discussion.

**Conflicts of Interest:** The authors declare no conflict of interest.

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
