# Peer review of "3D Tumor Spheroid and Organoid to Model Tumor Microenvironment for Cancer Immunotherapy"

_2674-1172, doi:10.3390/organoids1020012_

Round 1

Reviewer 1 Report

The manuscript by Yichen Zhu et al. entitled “3D tumor spheroid in model tumor microenvironment for cancer therapy.” presents the application of tumor spheroid and organoid models to cancer immunotherapy. The review is well structured.

In the present manuscript, the authors presented spheroid models while they described these models have their limitations.

In my opinion, the title of this manuscript should be cleared by including organoids and immunotherapy. In addition, the authors will improve the abstract by considering this point.

Figure 1b represents the hypoxia model, but the authors need to modify the figure to show a change of immune cells in a hypoxic tumor. In Figure 2. tissue processing is usually a general method. The authors need to improve the figure by deleting it.

I recommend the authors include a table or figure to compare the advantage and disadvantages between spheroid and organoid.

Author Response

Major comments

  1. In my opinion, the title of this manuscript should be cleared by including organoids and immunotherapy. In addition, the authors will improve the abstract by considering this point.

Response: We appreciate the Reviewer’s comments, and we have updated the title and abstract.

  1. Figure 1b represents the hypoxia model, but the authors need to modify the figure to show a change of immune cells in a hypoxic tumor. In Figure 2. tissue processing is usually a general method. The authors need to improve the figure by deleting it.

Response: We appreciate the Reviewer’s comments, and we have updated Figure 1b. For Figure 2, we also deleted the Tissue Processing section.

  1. I recommend the authors include a table or figure to compare the advantage and disadvantages between spheroid and organoid.

Response: We appreciate the Reviewer’s comments, and we have generated a table comparing spheroids and organoids.

Reviewer 2 Report

The paper “3D Tumor Spheroid to Model Tumor Microenvironment for Cancer Therapy” by Zhu et al reviews the 3D spheroid culture to mimic TME for developing cancer therapeutics. The paper introduces a lot of interesting articles, yet there are several issues. It is recommended the article should be significantly revised before further consideration. Detailed information is listed below:

1.     The overall organization of the article should be improved. The relationship between sections is confusing. It is recommended that the authors revisit titles of sections 2, 3, 4, and 5 and re-organize the article.

2.     It is recommended that the authors can include some summary tables in the article, so the audience might be easy to follow.

3.     While the authors claim “For instance, the most widely used fluorescence microscopy method, confocal microscopy, has a low penetration depth that prevents it from imaging large tumor spheroids,” there are actually articles estimating the viability of 3D Cancer spheroids in a label-free manner and also machine learning. It is recommended that the authors can include those.

4.     Microfluidics is an important player in 3D cell culture to generate uniform spheroids. The authors mentioned that, but it is overall downplayed. It is recommended that the authors can better highlight relevant technology in a section/sub-section.

Author Response

  1. The overall organization of the article should be improved. The relationship between sections is confusing. It is recommended that the authors revisit titles of sections 2, 3, 4, and 5 and re-organize the article.

Response: We appreciate the Reviewer’s comments. We have renamed the sections to “1. Introduction: The Tumor Microenvironment as a Barrier to Cancer Immunotherapy,” “2. 3D modeling of the TME,” “3. Cell Line-Derived Tumor Spheroids,” “3.1. Formation Methods”“3.2. TME Modeling Capabilities of Spheroids,” and “4. Patient-Derived Spheroid and Organoid Models.” We have also included relevant sub-sections to further clarify the structure of the article and have reorganized the text to better fit the new structure.

  1. It is recommended that the authors can include some summary tables in the article, so the audience might be easy to follow.

Response: We appreciate the Reviewer’s comments, and we have generated a table comparing spheroid and organoid, which is a comprehensive summary of our main focus

  1. While the authors claim “For instance, the most widely used fluorescence microscopy method, confocal microscopy, has a low penetration depth that prevents it from imaging large tumor spheroids,” there are actually articles estimating the viability of 3D Cancer spheroids in a label-free manner and also machine learning. It is recommended that the authors can include those.

Response: We appreciate the Reviewer’s insight onto this issue. We have added these sentences to the Conclusionsection to address these technologies (See Page 12):

“However, this could be mitigated by recent advances in various noninvasive imaging methods such as electrical impedance, which has the ability to image larger spheroids using their electrical properties (139-142). Analysis of spheroid characteristics such as viability and invasiveness could also be expediated with integration of automation and artificial intelligence”

  1. Microfluidics is an important player in 3D cell culture to generate uniform spheroids. The authors mentioned that, but it is overall downplayed. It is recommended that the authors can better highlight relevant technology in a section/sub-section.

Response: We appreciate the Reviewer’s comments, and we added a paragraph in Section 3.1. Formation Methodsabout the scaffold-free methods, including microfluidics. Since this is one type of formation methods in cell-line derived spheroid, we prefer not to add an entire section (See Page 5).

“Besides scaffolding, another four approaches have been utilized for spheroid formation in a scaffold-free culture (140). In agitation-based methods, cells are kept in stirring condition to avoid cell adhesion to surfaces for enhanced spheroid aggregation (140). The hanging drop technique grows spheroids within drops of culture medium, taking advantage of the surface tension of liquid (140, 141). In the liquid overlay technique, non-adhesive surfaces such as agarose are seeded with cells for spheroid formation without cell attachment (142). In contrast, the microfluidic system, has cell culture microchambers and hemispherical microwellswith a concentration gradient generator(143). The complex system provides controlled mixing, chemical concentration gradients, lower reagent consumption, continuous perfusion and precise control of pressure and shear stress on cells (144). It was reported by Ruppen et al. that spheroid formed with continuous perfusion of drugs would have higher drug resistivity (145). With some further enhancement, long-term monitoring and high-throughput drug screening platforms could also been achieved through the microfluidic system (143, 146).”

Round 2

Reviewer 2 Report

The authors addressed the comments from the reviewers.